# Thermodynamics of the Coarse-Graining Master Equation

**DOI:** 10.3390/e22050525

**Published:** 2020-05-05

**Authors:** Gernot Schaller, Julian Ablaßmayer

**Affiliations:** Institut für Theoretische Physik, Technische Universität Berlin, Hardenbergstr. 36, 10623 Berlin, Germany; julian.ablassmayer@gmail.com

**Keywords:** master equation, full counting statistics, secular approximation, entropy production

## Abstract

We study the coarse-graining approach to derive a generator for the evolution of an open quantum system over a finite time interval. The approach does not require a secular approximation but nevertheless generally leads to a Lindblad–Gorini–Kossakowski–Sudarshan generator. By combining the formalism with full counting statistics, we can demonstrate a consistent thermodynamic framework, once the switching work required for the coupling and decoupling with the reservoir is included. Particularly, we can write the second law in standard form, with the only difference that heat currents must be defined with respect to the reservoir. We exemplify our findings with simple but pedagogical examples.

## 1. Introduction

With the advent of an era where the promises of quantum computation [1] are approached in laboratories, one has to face the problem that controlled quantum systems are inevitably coupled to the outside world. The outside world can be approximated as a reservoir, which by construction contains infinitely many degrees of freedom. Since the required resources to simulate even finite quantum systems on classical computers scale exponentially with the number of constituents, the exact solution of the system-reservoir dynamics is futile except from a few exactly solvable cases.

Therefore, one typically aims to describe the dynamical evolution of an open quantum system by means of its reduced (system) density matrix only. To preserve the probability interpretation, the dynamical map governing the time evolution of the reduced density matrix should preserve its fundamental properties like trace, hermiticity, and positive semidefiniteness, at least in an approximate sense. While it is known that the exact dynamical map can be represented as a Kraus map [2] with intriguing mathematical properties [3], such a Kraus map is in general difficult to obtain from microscopic parameters. Many authors thereby follow the approach to find a first order differential equation with constant coefficients for the system density matrix. Here, the Lindblad–Gorini–Kossakowski–Sudarshan (LGKS) form master equation [4,5] stands out as it always preserves the density matrix properties. Although only a small fraction of Kraus maps can be represented as exponentiated LGKS generators [6], the class of LGKS generators is important since there exist standard routes to obtain them via microscopic derivations [7,8,9,10] from a global Hamiltonian of system and reservoir and their interaction. Technically, the standard route [8] is built on three basic assumptions: First, the Born approximation involves at least initially a factorization assumption between system and reservoir. Second, the Markovian approximation assumes that the reservoir re-equilibrates much faster than the system. Together these two in general suffice to obtain a time-independent generator that preserves trace and hermiticity. Third, to obtain a generator of LGKS form, it is additionally necessary to apply the secular approximation that assumes that the splitting between system energies is large. For a reservoir in thermal equilibrium, the dynamical map obtained this way will drag the system density matrix towards the local thermal equilibrium state of the system (which does not depend on the system-reservoir coupling characteristics) and moreover has a transparent thermodynamic interpretation [11,12,13]. This has sparked ideas to explore the potential of open quantum systems as quantum heat engines [14], which is nowadays part of a somewhat larger research field called quantum thermodynamics [15]. Clearly, the approximations required to arrive at a LGKS generator may become invalid for realistic systems, and it has, e.g., been highlighted that the use of LGKS generators may lead to inaccuracies [16] and even unphysical artifacts such as finite currents through disconnected regions or discontinuous dependence on parameters [17]. These shortcomings need not be taken as argument against LGKS approaches in general [18] but should be considered as a warning to mind the region of validity and as a motivation to develop alternative derivation schemes with controlled approximations [19,20,21].

In this paper, we will consider the coarse-graining approach [18,22,23,24,25,26,27,28,29,30], which by construction for short coarse-graining times approaches the exact short-time dynamics, is always of LGKS form, and for large coarse-graining times performs a secular approximation. For fixed coarse-graining times it effectively implements a partial secular approximation [31,32], for which to best of our knowledge a thermodynamic interpretation has only been performed from the system perspective [30] without an exact assessment of the reservoir heat.

The article is organized as follows: In Section 2 we introduce the coarse-graining generator with a counting field resolving the energy changes of the reservoir and discuss its properties. In Section 3 we then discuss the energy conservation and show a second-law type inequality for the entropy production rate. We proceed by exemplifying this for simple model systems where analytic approaches are possible like the spin-boson pure-dephasing model in Section 4, the single resonant level in Section 5, and the single-electron transistor in Section 6, before concluding in Section 7. Technical derivations are provided in an appendix.

## 2. Fixed-Time Coarse-Graining

We consider an open quantum system that is (possibly repeatedly) brought into contact with a unit of a stream of reservoirs as depicted in Figure 1.

In contrast to collisional models [33,34,35,36], we consider each unit of the reservoir stream to be of infinite size, such that even for a single system-unit interaction, the dynamics cannot be solved in general.

The total Hamiltonian of our setup can be written as
(1)$$\begin{array}{c} H\left(t\right)=H_{S}+\sum_{n}g_{n}\left(t\right)H_{I,n}+\sum_{n}H_{B,n} \end{array}$$
with system Hamiltonian $H_{S}$ and reservoir Hamiltonian $H_{B,n}$ of unit *n*. The dimensionless coupling functions $g_{n}\left(t\right)$ sequentially turn on and off the interaction. For simplicity, we will consider them as piecewise constant and non-overlapping $g_{n}\left(t\right)g_{n+1}\left(t+0^{+}\right)=0$, but these conditions may be somewhat relaxed. Further, each interaction Hamiltonian can be expanded in terms of system ($A_{\alpha }$) and reservoir ($B_{\alpha }$) coupling operators
(2)$$\begin{array}{c} H_{I,n}=\sum_{\alpha }A_{\alpha }⊗B_{\alpha ,n}=H_{I,n}^{†}\;. \end{array}$$

Although achievable by suitable transformations, we do in this paper not require that system and bath coupling operators are individually hermitian. At the beginning of the interaction, each reservoir unit is prepared in the state $\rho_{B,n}$. Since in the following we assume that all the reservoir units are identically prepared and coupled $\rho_{B,n}\hat{=}\rho_{B}$, $B_{\alpha ,n}\hat{=}B_{\alpha }$, $H_{B,n}\hat{=}H_{B}$, $H_{I,n}\hat{=}H_{I}$, we will drop the index *n*, which just served as a reminder on which Hilbert space the associated operators are acting. This setting can be easily generalized to multiple reservoirs that are coupled simultaneously—in Figure 1 these would just induce parallel streams. To probe the case of just a constant system-reservoir interaction, we may consider the case τ→∞, where the standard weak-coupling thermodynamic analysis applies.

For practical calculations, a time-local first order differential equation for the system density matrix—a master equation—is beneficial, since it allows for a simple propagation of the system density matrix. Going to the interaction picture (bold symbols) with respect to $H_{S}+H_{B}$ allows to microscopically derive such a LGKS generator. Specifically, it follows by the demand to find a time-local generator $L_{\tau }$ for the system that yields the same dynamics as the exact solution after coarse-graining time τ
(3)$$\begin{array}{c} \exp\left\{L_{\tau }\cdot \tau \right\}\rho_{S}\left(t_{0}\right)={\mathrm{Tr}}_{B}\left\{U\left(t_{0}+\tau ,t_{0}\right)\rho_{S}\left(t_{0}\right)⊗\rho_{B}U^{†}\left(t_{0}+\tau ,t_{0}\right)\right\} \end{array}$$
for all initial states of the system $\rho_{S}\left(t_{0}\right)$ and identical initial reservoir states $\rho_{B}$. Consistently, we have also dropped the index *n* in the dissipator (calligraphic symbols denote superoperators throughout). The r.h.s. can be determined perturbatively [22], which allows to explicitly calculate the generator of the evolution. Since we generalize the setting by allowing for initial times $t_{0}>0$, we detail this derivation in Appendix A. Additionally, to track the statistics of energy entering the reservoir unit *n*, the dissipator can be generalized by a counting field, and a microscopic derivation of this along the lines of Ref. [37] is provided in Appendix B. The generalized coarse-graining master equation is then given by $\frac{d}{dt}\rho_{S}=L_{\tau }\left(\xi \right)\rho_{S}$ with
(4)$$\begin{array}{cc} L_{\tau }\left(\xi \right)\rho_{S} & =-i\left[\frac{1}{2i\tau }\sum_{\alpha \beta }\overset{t_{0}+\tau }{\underset{t_{0}}{\;\int \int \;}}dt_{1}dt_{2}C_{\alpha \beta }\left(t_{1}-t_{2}\right)\mathrm{sgn}\left(t_{1}-t_{2}\right)A_{\alpha }\left(t_{1}\right)A_{\beta }\left(t_{2}\right),\rho_{S}\right]\\ \; & \;+\frac{1}{\tau }\sum_{\alpha \beta }\overset{t_{0}+\tau }{\underset{t_{0}}{\;\int \int \;}}dt_{1}dt_{2}[C_{\alpha \beta }^{\xi }\left(t_{1}-t_{2}\right)A_{\beta }\left(t_{2}\right)\rho_{S}A_{\alpha }\left(t_{1}\right)\\ \; & \;\;-\frac{C_{\alpha \beta }\left(t_{1}-t_{2}\right)}{2}\left\{A_{\alpha }\left(t_{1}\right)A_{\beta }\left(t_{2}\right),\rho_{S}\right\}]\;. \end{array}$$
Here, $A_{\alpha }\left(t\right)=e^{+iH_{S}t}A_{\alpha }e^{-iH_{S}t}$ are the system coupling operators in the interaction picture (bold symbols throughout), and the generalized reservoir correlation functions
(5)$$\begin{array}{cc} C_{\alpha \beta }^{\xi }\left(t_{1}-t_{2}\right) & =\mathrm{Tr}\left\{e^{-iH_{B}\xi }e^{+iH_{B}\left(t_{1}-t_{2}\right)}B_{\alpha }e^{-iH_{B}\left(t_{1}-t_{2}\right)}e^{+iH_{B}\xi }B_{\beta }\rho_{B}\right\}\;,\\ C_{\alpha \beta }\left(t_{1}-t_{2}\right) & \equiv C_{\alpha \beta }^{0}\left(t_{1}-t_{2}\right) \end{array}$$
encode the reservoir properties, where we use an (initial) grand-canonical equilibrium state
(6)$$\begin{array}{c} \rho_{B}=\frac{e^{-\beta (H_{B}-\mu N_{B})}}{Z_{B}} \end{array}$$
with inverse temperature β, chemical potential μ, and partition function $Z_{B}=\mathrm{Tr}\left\{e^{-\beta (H_{B}-\mu N_{B})}\right\}$. In case of multiple reservoirs that are simultaneously coupled to the system, this is generalized to a tensor product of local equilibrium states. The superoperator $L_{\tau }$ evidently also depends on $t_{0}$ and the reservoir properties, which for the sake of brevity we do not make explicit. We summarize a few useful properties of Equation (Equation 4):For ξ=0 the conventional fixed-time coarse-graining master equation [22,23] is reproduced. Notationally, we will denote this limit as $L_{\tau }\equiv L_{\tau }\left(0\right)$. Previous studies (for $t_{0}=0$ [22]) have shown that $L_{\tau }$ is always of LGKS form, and we can also confirm this for finite $t_{0}$, see Appendix C. Thus, Spohn’s inequality [12]
(7)$$\begin{array}{c} \sigma_{\tau }\equiv -\mathrm{Tr}\left\{\left(L_{\tau }\rho_{S}\left(t\right)\right)\left[\ln\rho_{S}\left(t\right)-\ln{\bar{\rho }}_{\tau }\right]\right\}\geq 0 \end{array}$$
holds with any nonequilibrium steady state ${\bar{\rho }}_{\tau }$ obeying $L_{\tau }{\bar{\rho }}_{\tau }=0$ (which may in general depend on $t_{0}$ as well).It has been debated whether local or global LGKS approaches are more suitable to discuss quantum thermodynamics [34,38,39,40]. To see how the dissipator (Equation 4) locates in this discussion, let us assume that our system is composed of multiple subsystems that are coupled by some constant interaction. Then, system coupling operators $A_{\alpha }$ that in the Schrödinger picture act locally on a subsystem component will in general transfer to non-local interaction-picture operators $A_{\alpha }\left(t\right)$. Thereby, the Lindblad operators from the LGKS generator (Equation 4) will in general globally act on the whole system. An obvious exception arises in the case when the time-dependence of the system operators itself is negligible $A_{\alpha }\left(t\right)\approx A_{\alpha }$, which happens, e.g., in the singular coupling limit [8] or for very short coarse-graining times. Another exception arises when the couplings between the subsystem components are comparably weak, such that the operators in the interaction picture $A_{\alpha }\left(t\right)$ remain approximately local over the course of the coarse-graining timescale τ.By going to the energy eigenbasis of the system, it is possible to cast the dissipator (Equation 4) into a single-integral form. Furthermore, for τ→∞, the Born-Markov-secular (BMS) master equation [8] is reproduced [10]
(8)$$\begin{array}{c} \lim_{\tau \to \infty }L_{\tau }=L_{\mathrm{BMS}}\;, \end{array}$$
such that the secular approximation can be performed by τ→∞, which we detail also for finite $t_{0}$ in Appendix D. We also find that in the secular limit, the energy current entering the system and the energy current leaving the reservoir are identical, which demonstrates that a secular approximation imposes energy conservation between system and reservoir.When the dissipator does not depend on the initial time $t_{0}$—this happens, e.g., when only certain combinations of coupling operators contribute $A_{\alpha }\left(t_{1}\right)=A_{\alpha }e^{+i\epsilon_{\alpha }t_{1}}$ and $A_{\beta }\left(t_{2}\right)=A_{\alpha }^{†}e^{-i\epsilon_{\alpha }t_{2}}$ such that the integrand in Equation (Equation 4) depends only on $t_{1}-t_{2}$— the system will under repeated system-reservoir couplings relax to the nonequilibrium steady state ${\bar{\rho }}_{\tau }$. When this nonequilibrium steady state is reached, Spohn’s inequality (Equation 7) would predict a vanishing entropy production rate.

As we will show in this paper, despite the fact that ${\bar{\rho }}_{\tau }$ is a nonequilibrium steady state already for a single stream of reservoirs, a thermodynamic interpretation of the coarse-graining master equation for finite $t_{0}$ and τ is possible. The conservation of energy then requires to take into account the work required for coupling and decoupling the reservoir and one can then demonstrate positivity of a global entropy production rate, which involves system and reservoir units altogether.

## 3. Thermodynamics

### 3.1. Energetic Balance

The energy change of system and reservoir together must be balanced by the switching work spent to couple them via $g_{n}\left(t\right)$ at $t_{0}$ and to decouple them at $t_{0}+\tau$
(9)$$\begin{array}{c} \Delta E_{S}\left(t_{0}+\tau ,t_{0}\right)+\Delta E_{B}\left(t_{0}+\tau ,t_{0}\right)=\Delta W\left(t_{0}+\tau ,t_{0}\right)\;. \end{array}$$
Thus, when the switching work is negligible, this implies that the energetic changes in the reservoir can be deduced from the changes in the system, and an explicit counting field analysis would not be necessary. To the contrary, when the system density matrix has reached a (possibly stroboscopic) steady state such that $\Delta E_{S}$ can be neglected, all the switching work invested is dissipated as heat into the reservoir. It is reassuring to test the energy conservation explicitly, see Appendix E.

Since we will in general not be able to write down exact expressions for the energetic system and reservoir changes and the switching work, we in the following derive expressions based on (Equation 4) valid to second order in the system-reservoir interaction strength. For fixed coarse-graining time τ, the time-dependent solution of the coarse-graining master equation is given by $\rho_{S}\left(t\right)=e^{L_{\tau }\left(t-t_{0}\right)}\rho_{S}^{0}$. By using this dissipator, we will of course match the initial condition $\rho_{S}^{0}$ when $t=t_{0}$. Likewise, for $t-t_{0}=\tau$, one will best approximate the true solution. Using the dissipator $L_{\tau }$ for times $0<t-t_{0}<\tau$ just yields coarse-grained estimates of the evolution while the system is in contact with the first unit. Whereas for $t-t_{0}=n\tau$ with n∈N, the solution describes *n* successive interactions with units, the choice $n\tau <t-t_{0}<\left(n+1\right)\tau$ yields coarse-grained estimates as solution for *n* successive interactions that have passed while the system is in the process of interacting with the n+1st unit. Thus, t>0 can be chosen freely while τ is fixed. Then, the system energy change is for fixed τ just given by
(10)$$\begin{array}{c} I_{E,S}\left(t\right)\equiv \frac{d}{dt}\Delta E_{S}\left(t,t_{0}\right)=\mathrm{Tr}\left\{H_{S}\left(L_{\tau }e^{L_{\tau }\left(t-t_{0}\right)}\rho_{S}^{0}\right)\right\}=\mathrm{Tr}\left\{H_{S}\left(L_{\tau }\rho_{S}\left(t\right)\right)\right\}\;. \end{array}$$
We denote this as energy current entering the system, adopting the convention that positive contributions increase the system energy. Furthermore, for an additive decomposition of the dissipator into multiple reservoir contributions $L_{\tau }\to \sum_{\nu }L_{\tau }^{\nu }$ it is straightforward to also decompose the current into contributions entering from reservoir ν
(11)$$\begin{array}{c} I_{E,S}^{\left(\nu \right)}\left(t\right)\equiv \mathrm{Tr}\left\{H_{S}\left(L_{\tau }^{\nu }\rho_{S}\left(t\right)\right)\right\}\;. \end{array}$$

To obtain the energy change of the reservoir, we consider the counting field ξ. The first moment of the energy change can be computed by the first derivative with respect to the counting field (see Appendix B), whereas for the energy current we consider an additional time derivative. Then, we have for the energy current leaving the reservoir (this is positive when it decreases the reservoir energy)
(12)$$\begin{array}{cc} I_{E,B}\left(t\right) & \equiv -\frac{d}{dt}\Delta E_{B}\left(t,t_{0}\right)=+i\partial_{\xi }\mathrm{Tr}\left\{L_{\tau }\left(\xi \right)e^{L_{\tau }\left(\xi \right)\left(t-t_{0}\right)}\rho_{S}\left(t_{0}\right)\right\}{|}_{\xi =0}\\ \; & =\mathrm{Tr}\left\{\left(+i\partial_{\xi }L_{\tau }{\left(\xi \right)|}_{\xi =0}\right)\rho_{S}\left(t\right)\right\}\;, \end{array}$$
where we have used the trace conservation property of $L_{\tau }$. Furthermore, here, for multiple reservoirs, an additive decomposition of the dissipator $L_{\tau }\left(\xi \right)\to \sum_{\nu }L_{\tau }^{\nu }\left(\xi_{\nu }\right)$ with reservoir-specific counting field $\xi_{\nu }$ transfers to an additive decomposition of the total current
(13)$$\begin{array}{cc} I_{E,B}^{\left(\nu \right)}\left(t\right) & \equiv \mathrm{Tr}\left\{\left(+i\partial_{\xi_{\nu }}L_{\tau }^{\nu }\left(\xi_{\nu }\right){|}_{\xi_{\nu }=0}\right)\rho_{S}\left(t\right)\right\}\;. \end{array}$$

In the secular limit τ→∞ we can show that the currents in Equations (Equation 10) and (Equation 12) coincide, see Appendix D. By the conservation of energy (Equation 9), we therefore define the switching power as
(14)$$\begin{array}{c} P_{\mathrm{sw}}\left(t\right)\equiv \sum_{\nu }I_{E,S}^{\left(\nu \right)}\left(t\right)-I_{E,B}^{\left(\nu \right)}\left(t\right)\;, \end{array}$$
but in Appendix E we also provide an independent approximation (Equation 76) to the switching work.

### 3.2. Entropic Balance

We start from the generalized coarse-graining master Equation (Equation 4) with an energy counting field ξ. We can evaluate this equation in the (orthonormal) basis (see also, e.g., [41]) where its solution, the time-dependent density matrix, is diagonal
(15)$$\begin{array}{c} \rho_{S}\left(t\right)=\sum_{j}P_{j}\left(t\right)|j(t)\rangle \langle j(t)|\;, \end{array}$$
such that $|j(t)\rangle$ represent the eigenstates and $P_{j}\left(t\right)$ the eigenvalues of the density matrix. Only when τ→∞ and the system relaxes to a steady state, this would correspond to the system energy eigenbasis (see Appendix D), but in general this basis will be different. To describe also models with particle exchange between system and reservoir, we additionally assume that these eigenstates are also eigenstates of the system particle number operator $N_{S}|j(t)\rangle =N_{j}|j(t)\rangle$, i.e., the system density matrix must not contain superpositions of states with different particle numbers. Then, by evaluating Equation (Equation 4) in the basis $|j(t)\rangle$, one finds that the eigenvalues $P_{i}\left(t\right)$ obey a generalized rate equation
(16)$$\begin{array}{c} \frac{d}{dt}P_{i}\left(t\right)=\sum_{j}R_{ij}^{\tau }\left(\xi \right)P_{j}\left(t\right)-\sum_{j}R_{ji}^{\tau }\left(0\right)P_{i}\left(t\right)\;, \end{array}$$
and the (tacitly time-dependent) transition rate from j→i is generated by the jump term of Equation (Equation 4)
(17)$$\begin{array}{cc} R_{ij}^{\tau }\left(\xi \right) & =\langle i(t)|\left(L_{\tau }\left(\xi \right)|j(t)\rangle \langle j(t)|\right)|i(t)\rangle \\ \; & =\frac{1}{\tau }\overset{t_{0}+\tau }{\underset{t_{0}}{\;\int \int \;}}dt_{1}dt_{2}\sum_{\alpha \beta }C_{\alpha \beta }\left(t_{1}-t_{2}-\xi \right)\langle i|A_{\beta }\left(t_{2}\right)|j\rangle \langle j|A_{\alpha }\left(t_{1}\right)|i\rangle \\ \; & =\int d\omega \sum_{\alpha \beta }\gamma_{\alpha \beta }\left(\omega \right)e^{+i\omega \xi }\frac{1}{2\pi \tau }\overset{t_{0}+\tau }{\underset{t_{0}}{\;\int \int \;}}dt_{1}dt_{2}e^{-i\omega (t_{1}-t_{2})}\langle i|A_{\beta }\left(t_{2}\right)|j\rangle \langle j|A_{\alpha }\left(t_{1}\right)|i\rangle \\ \; & \equiv \int R_{ij,+\omega }^{\tau }e^{+i\omega \xi }d\omega \;, \end{array}$$
where we have omitted the time-dependence of the eigenstates for brevity. The energy-resolved quantity $R_{ij,+\omega }^{\tau }$ is thus also time-dependent but unambiguously defined by a Fourier transform with respect to the counting field. In Appendix F we detail that $R_{ij,\omega }^{\tau }\geq 0$ and hence it can be interpreted as a rate for processes with a system transition from j→i that go along with a reservoir energy change +ω. In the eigenbasis of the time-dependent solution $\rho_{S}\left(t\right)$, the energy current leaving the reservoir (Equation 12) can be represented in the standard rate equation form (albeit with time-dependent rates)
(18)$$\begin{array}{cc} I_{E,B}\left(t\right) & =-\int d\omega \omega \sum_{ij}R_{ij,+\omega }^{\tau }P_{j}\left(t\right)\;. \end{array}$$
However, we emphasize that the above current can be determined using Equation (Equation 12) without diagonalizing the time-dependent density matrix. For the specific examples we consider below, even an analytic calculation of the time-dependent rates $R_{ij,+\omega }^{\tau }$ is possible. When the total particle number (of system and reservoir unit together) is conserved $\left[H_{S},N_{S}\right]=\left[H_{B},N_{B}\right]=\left[H_{I},N_{S}+N_{B}\right]=0$, any particle change $N_{i}-N_{j}$ in the system is accompanied by the corresponding negative change $N_{j}-N_{i}$ in the reservoir, such that a matter current leaving the reservoir or entering the system can be defined in analogy to Equation (Equation 10)
(19)$$\begin{array}{cc} I_{M,S}\left(t\right)=I_{M,B}\left(t\right) & =\mathrm{Tr}\left\{N_{S}L_{\tau }\rho_{S}\left(t\right)\right\}=\sum_{ij}\left(N_{i}-N_{j}\right)R_{ij}^{\tau }\left(0\right)P_{j}\left(t\right)\;. \end{array}$$
We then show in Appendix F that the energy-resolved time-dependent rates obey a detailed balance relation
(20)$$\begin{array}{cc} \frac{R_{ij,+\omega }^{\tau }}{R_{ji,-\omega }^{\tau }} & =e^{+\beta [\omega -\mu \left(N_{j}-N_{i}\right)]}\;, \end{array}$$
whereas the integrated rates $R_{ij}^{\tau }\left(0\right)$ do not. For multiple reservoirs characterized by local equilibrium states of inverse temperature $\beta_{\nu }$ and chemical potential $\mu_{\nu }$, each fulfilling $\mathrm{Tr}\left\{H_{I}\rho_{B}^{\left(\nu \right)}\right\}=0$, we have under the weak-coupling assumption an additive decomposition of rates
(21)$$\begin{array}{c} R_{ij,+\omega }^{\tau }=\sum_{\nu }R_{ij,+\omega }^{\tau ,(\nu )}\;, \end{array}$$
where $R_{ij,+\omega }^{\tau ,(\nu )}$ represents the individual contribution of the νth reservoir. Then, the detailed balance relation (Equation 20) holds locally and also the matter current can be written in a reservoir-specific form
(22)$$\begin{array}{cc} I_{M,B}^{\left(\nu \right)}\left(t\right) & \equiv \mathrm{Tr}\left\{N_{S}L_{\tau }^{\nu }\rho_{S}\left(t\right)\right\}\;. \end{array}$$
We show in Appendix F that the second law can with (Equation 22) and (Equation 13) be written as
(23)$$\begin{array}{c} {\dot{S}}_{i}^{\tau }=\dot{S}-\sum_{\nu }\beta_{\nu }\left(I_{E,B}^{\left(\nu \right)}\left(t\right)-\mu_{\nu }I_{M,B}^{\left(\nu \right)}\left(t\right)\right)\geq 0\;, \end{array}$$
where $S=-\mathrm{Tr}\left\{\rho_{S}\left(t\right)\ln\rho_{S}\left(t\right)\right\}$ is the entropy of the system only and the other terms describe the entropy produced in the reservoir units. Individually, each of these contributions may become negative and is only subject to the constraint that the second law is obeyed globally.

We have constrained ourselves to fixed coarse-graining times, for which we can write the second law in differential form, since the usual LGKS formalism, albeit with differently defined energy currents, applies. Considering the dynamical coarse-graining approach [23,28,30], we note that the integrated entropy production $\Delta_{i}S\left(\tau \right)=\int_{0}^{\tau }{\dot{S}}_{i}^{\tau }\left(t\right)dt\geq 0$ is then evidently also positive but not necessarily a monotonously growing function of τ.

## 4. Example: Pure-Dephasing Spin-Boson Model

### 4.1. Model and Exact Results

The pure dephasing spin-boson model describes a two-level system
(24)$$\begin{array}{c} H_{S}=\frac{\omega }{2}\sigma^{z} \end{array}$$
with energy splitting ω that is coupled via a purely dephasing interaction
(25)$$\begin{array}{c} H_{I}=\sigma^{z}⊗\sum_{k}\left(h_{k}b_{k}+h_{k}^{*}b_{k}^{†}\right) \end{array}$$
with spontaneous emission amplitudes $h_{k}$ to a reservoir of harmonic oscillators
(26)$$\begin{array}{c} H_{B}=\sum_{k}\omega_{k}b_{k}^{†}b_{k} \end{array}$$
of (positive) energies $\omega_{k}$.

Since interaction and system Hamiltonian commute, the model can be solved exactly [22,42], and from the exact solution one finds that the populations in the system energy eigenbasis remain constant, whereas the coherences decay
(27)$$\begin{array}{c} \langle 0|\rho_{S}^{\mathrm{ex}}\left(t\right)|1\rangle =\exp\left\{-\frac{4}{\pi }\int_{0}^{\infty }\Gamma \left(\omega \right)\frac{\sin^{2}\left(\omega t/2\right)}{\omega^{2}}\coth\left(\frac{\beta \omega }{2}\right)d\omega \right\}\rho_{01}^{0}\;, \end{array}$$
where β denotes the inverse reservoir temperature (we consider μ=0) and $\Gamma \left(\omega \right)=2\pi \sum_{k}{\left|h_{k}\right|}^{2}\delta \left(\omega -\omega_{k}\right)$ the spectral density of the reservoir. Constant populations in the system eigenbasis imply that the system energy remains constant. Additionally, the exact solution also predicts that energy is radiated into the reservoir (see Appendix G)
(28)$$\begin{array}{c} \Delta E_{B}^{\mathrm{ex}}\left(t,0\right)=\frac{2}{\pi }\int_{0}^{\infty }\frac{\Gamma (\omega )}{\omega }\sin^{2}\left(\frac{\omega t}{2}\right)d\omega \;, \end{array}$$
which does not depend on initial system and reservoir states and stems from the interaction Hamiltonian.

### 4.2. Coarse-Graining Dynamics

The coarse-graining dissipator (Equation 4) for the pure-dephasing model is particularly simple as the system coupling operator in the interaction picture carries no time-dependence
(29)$$\begin{array}{cc} L_{\tau }\left(\xi \right)\rho_{S} & =\frac{1}{\tau }\overset{t_{0}+\tau }{\underset{t_{0}}{\;\int \int \;}}dt_{1}dt_{2}\left[C\left(t_{1}-t_{2}-\xi \right)\sigma^{z}\rho_{S}\sigma^{z}-C\left(t_{1}-t_{2}\right)\rho_{S}\right]\;. \end{array}$$
Here, the Lamb-shift contribution has been dropped as it is proportional to the identity, and the correlation function is given by $C\left(\Delta t\right)=\frac{1}{2\pi }\int d\omega \bar{\Gamma }\left(\omega \right)\left[1+n_{B}\left(\omega \right)\right]e^{-i\omega \Delta t}$ where the analytic continuation of the spectral density as an odd function is understood $\bar{\Gamma }\left(-\left|\omega \right|\right)=-\Gamma \left(\left|\omega \right|\right)$ and $\bar{\Gamma }\left(+\left|\omega \right|\right)=+\Gamma \left(\left|\omega \right|\right)$, and $n_{B}\left(\omega \right)={\left[e^{\beta \omega }-1\right]}^{-1}$ denotes the Bose distribution. Since the integrand only depends on the difference $t_{1}-t_{2}$, the dissipator does not even depend on the initial time $t_{0}$. The solution of the above differential equation predicts a decay of coherences ($t_{0}=0$)
(30)$$\begin{array}{c} \langle 0|\rho_{S}\left(t\right)|1\rangle =\exp\left\{-2\int \bar{\Gamma }\left(\omega \right)\left[1+n_{B}\left(\omega \right)\right]\frac{\tau }{2\pi }{\mathrm{sinc}}^{2}\left[\frac{\omega \tau }{2}\right]d\omega \cdot t\right\}\langle 0|\rho_{S}\left(0\right)|1\rangle \;, \end{array}$$
where sinc(x)≡sin(x)/x, whereas populations remain constant. For t=τ, this result matches the exact solution (Equation 27) [23], i.e., we have $\rho_{S}^{\mathrm{exact}}\left(t\right)=e^{L_{t}\cdot t}\rho_{S}^{0}$. The equivalence of (Equation 30) and (Equation 27) can be explicitly seen by rewriting in the above equation the negative frequency component of the integral.

### 4.3. Energetic Balance

Additionally, we show that also the energy radiated into the reservoir is faithfully reproduced by the generalized coarse-graining master equation. The energy current entering the system (Equation 10) vanishes
(31)$$\begin{array}{c} I_{E,S}\left(t\right)=0\;. \end{array}$$
We note that for this model, alternative constructions for a refined system energy current based on a time-dependent Hamiltonian of mean force would lead to the same result: Since the system Gibbs state $\rho_{\beta }=e^{-\beta H_{S}}/Z_{S}$ is just invariant under the pure dephasing dissipator, the ”refined heat flow” suggested in Equation (Equation 66) of Ref. [30] vanishes as well.

From the counting field formalism we do however obtain that the energy current leaving the reservoir (Equation 12) in this model remains finite and time-independent
(32)$$\begin{array}{cc} I_{E,B}\left(t\right)=I_{E,B} & =\frac{1}{\tau }\overset{t_{0}+\tau }{\underset{t_{0}}{\;\int \int \;}}dt_{1}dt_{2}{\left[i\partial_{\xi }C\left(t_{1}-t_{2}-\xi \right)\right]}_{\xi =0}\mathrm{Tr}\left\{\sigma^{z}\rho_{S}\left(t\right)\sigma^{z}\right\}\\ \; & =\frac{-1}{\tau }\int \frac{d\omega }{2\pi }\overset{t_{0}+\tau }{\underset{t_{0}}{\;\int \int \;}}dt_{1}dt_{2}\bar{\Gamma }\left(\omega \right)\left[1+n_{B}\left(\omega \right)\right]e^{-i\omega (t_{1}-t_{2})}\omega \\ \; & =-\int d\omega \omega \bar{\Gamma }\left(\omega \right)\left[1+n_{B}\left(\omega \right)\right]\frac{\tau }{2\pi }{\mathrm{sinc}}^{2}\left[\frac{\omega \tau }{2}\right]\\ \; & =-\int_{0}^{\infty }d\omega \omega \bar{\Gamma }\left(\omega \right)\frac{\tau }{2\pi }{\mathrm{sinc}}^{2}\left[\frac{\omega \tau }{2}\right]\;, \end{array}$$
where the last line can be shown by using $\bar{\Gamma }\left(-\omega \right)=-\bar{\Gamma }\left(\omega \right)$ and $\left[1+n_{B}\left(-\omega \right)\right]=-n_{B}\left(\omega \right)$ in the negative frequency components of the integral. The integral over this current over the coarse-graining time precisely matches the exact solution (Equation 28): $\tau I_{E,B}=\Delta E_{B}^{\mathrm{ex}}\left(\tau ,0\right)$. We also remark that the approximation to the switching work (Equation 76) becomes $\Delta W\left(t_{0}+\tau ,t_{0}\right)=-i\int_{t_{0}}^{t_{0}+\tau }dt_{1}\left[C\left(t_{1}-t_{0}-\tau \right)-C\left(t_{0}+\tau -t_{1}\right)\right]=\tau \int d\omega \bar{\Gamma }\left(\omega \right)\left[1+n_{B}\left(\omega \right)\right]\omega \frac{\tau }{2\pi }{\mathrm{sinc}}^{2}\left[\frac{\omega \tau }{2}\right]$, which in this case is exactly equivalent to the integral of the energy current leaving the reservoir (Equation 32) up to t=τ, i.e., to the end of the collision. Thus, in this model we obtain that the complete switching work is dissipated as heat into the reservoir $\Delta W\left(t_{0}+\tau ,t_{0}\right)=\Delta E_{B}\left(t_{0}+\tau ,t_{0}\right)$.

### 4.4. Entropic Balance

The energy-resolved rates become
(33)$$\begin{array}{c} R_{ij,+\omega }^{\tau }=\bar{\Gamma }\left(\omega \right)\left[1+n_{B}\left(\omega \right)\right]\frac{\tau }{2\pi }{\mathrm{sinc}}^{2}\left(\frac{\omega \tau }{2}\right){\left|\langle i(t)|\sigma^{z}|j(t)\rangle \right|}^{2}\;. \end{array}$$
With them, we can likewise confirm that the energy current (Equation 32) is time-independent—using completeness of the basis and ${\left(\sigma^{z}\right)}^{2}=1$. Inserting this in the global entropy production rate (Equation 23) we obtain
(34)$$\begin{array}{c} {\dot{S}}_{i}^{\tau }=\dot{S}+\beta \int_{0}^{\infty }d\omega \omega \bar{\Gamma }\left(\omega \right)\frac{\tau }{2\pi }{\mathrm{sinc}}^{2}\left[\frac{\omega \tau }{2}\right]\geq 0\;. \end{array}$$
This decomposes into two separately positive terms: The first term is the change of the system entropy, which by mere reduction of coherences just increases (see, e.g., [43]), and has been analyzed for this model before (see, e.g., [44]). Once the system has reached its stationary limit, it will vanish. The second term is positive since the integrand is positive but it remains finite for finite τ. Furthermore, since both the reduced system dynamics and the energy leaving the reservoir are exactly reproduced for t=τ, we also note that this matches the results of Ref. [45] when applied to the pure dephasing model.

We additionally remark that we can compare the global entropy production rate with the entropy production rate $\sigma_{\tau }$ based on Spohn’s inequality. Here, the second term in Equation (Equation 7) vanishes since $\ln{\bar{\rho }}_{S}$ has only diagonal and $L_{\tau }\rho_{S}$ has only off-diagonal components, such that $\sigma_{\tau }=\dot{S}$ yields only the entropy change in the system. Thus, in the pure-dephasing model, Spohn’s inequality completely neglects the entropy production in the reservoir, see also Figure 2 for a comparison.

## 5. Example: Single Resonant Level

### 5.1. Model

The single resonant level (SRL) is described by a single fermionic mode of energy ϵ (e.g., a quantum dot in the strong Coulomb blockade regime)
(35)$$\begin{array}{c} H_{S}=\epsilon d^{†}d \end{array}$$
that is tunnel-coupled to a fermionic reservoir with single-particle energies $\epsilon_{k}$
(36)$$\begin{array}{c} H_{B}=\sum_{k}\epsilon_{k}c_{k}^{†}c_{k} \end{array}$$
via the amplitudes $t_{k}$
(37)$$\begin{array}{c} H_{I}=d⊗\sum_{k}t_{k}c_{k}^{†}+d^{†}⊗\sum_{k}t_{k}^{*}c_{k}\;. \end{array}$$
Here, we have already represented the interaction Hamiltonian in terms of local system and reservoir fermions. Such a tensor product decomposition is possible using a Jordan-Wigner transform [17] but is typically performed tacitly. We can thus identify the system coupling operators $A_{1}\left(t\right)=d^{†}e^{+i\epsilon t}$ and $A_{2}\left(t\right)=de^{-i\epsilon t}$ and the Fourier transforms of the reservoir correlation functions $\gamma_{12}\left(\omega \right)=\Gamma \left(\omega \right)\left[1-f\left(\omega \right)\right]$ and $\gamma_{21}\left(\omega \right)=\Gamma \left(-\omega \right)f\left(-\omega \right)$ explicitly, where $\Gamma \left(\omega \right)=2\pi \sum_{k}{\left|t_{k}\right|}^{2}\delta \left(\omega -\epsilon_{k}\right)$ denotes the spectral density (also termed bare tunneling rate in this context) and $f\left(\omega \right)={\left[e^{\beta (\omega -\mu )}+1\right]}^{-1}$ the Fermi function of the reservoir in equilibrium. The model is also exactly solvable [24,46,47], but we will only consider the coarse-graining dynamics here (which converges to the exact solution, e.g., in the weak-coupling limit or for short times).

### 5.2. Coarse-Graining Dynamics

The coarse-graining master Equation (Equation 4) for the SRL reads in the interaction picture
(38)$$\begin{array}{cc} L_{\tau }\left(\xi \right)\rho_{S} & =\int d\omega \Gamma \left(\omega \right)\left[1-f\left(\omega \right)\right]\frac{\tau }{2\pi }{\mathrm{sinc}}^{2}\left[\frac{(\omega -\epsilon )\tau }{2}\right]\left[d\rho_{S}d^{†}e^{+i\omega \xi }-\frac{1}{2}\left\{d^{†}d,\rho_{S}\right\}\right]\\ \; & \;+\int d\omega \Gamma \left(\omega \right)f\left(\omega \right)\frac{\tau }{2\pi }{\mathrm{sinc}}^{2}\left[\frac{(\omega -\epsilon )\tau }{2}\right]\left[d^{†}\rho_{S}de^{-i\omega \xi }-\frac{1}{2}\left\{dd^{†},\rho_{S}\right\}\right]\;. \end{array}$$
It does not depend on $t_{0}$, since due to the structure of the correlation functions, only time differences enter Equation (Equation 4). An alternative motivation of such a dissipator with two terminals can be found via repeated projective measurements on the system that restore a product state between system and reservoir [48]. Further, since a single quantum dot does not carry any coherences, we have $\left[\rho_{S}\left(t\right),d^{†}d\right]=\left[\rho_{S}\left(t\right),dd^{†}\right]=0$, and the Lamb-shift type commutator term drops out from the beginning. Still, the dot populations can change under the dynamics. From the above dissipator, the probability of finding a filled dot follows the differential equation
(39)$$\begin{array}{c} \frac{d}{dt}P_{1}=\gamma_{\mathrm{in}}^{\tau }-\left(\gamma_{\mathrm{in}}^{\tau }+\gamma_{\mathrm{out}}^{\tau }\right)P_{1}\left(t\right)\;, \end{array}$$
with the positive rates
(40)$$\begin{array}{cc} \gamma_{\mathrm{in}}^{\tau } & =\int d\omega \Gamma \left(\omega \right)f\left(\omega \right)\frac{\tau }{2\pi }{\mathrm{sinc}}^{2}\left[\left(\omega -\epsilon \right)\tau /2\right]\;,\\ \gamma_{\mathrm{out}}^{\tau } & =\int d\omega \Gamma \left(\omega \right)\left[1-f\left(\omega \right)\right]\frac{\tau }{2\pi }{\mathrm{sinc}}^{2}\left[\left(\omega -\epsilon \right)\tau /2\right]\;, \end{array}$$
which for τ→∞ collapse to the usual secular description of the SRL. This differential equation can be readily solved
(41)$$\begin{array}{c} P_{1}\left(t\right)=\frac{\gamma_{\mathrm{in}}^{\tau }}{\gamma_{\mathrm{in}}^{\tau }+\gamma_{\mathrm{out}}^{\tau }}\left[1-e^{-\left(\gamma_{\mathrm{in}}^{\tau }+\gamma_{\mathrm{out}}^{\tau }\right)\left(t-t_{0}\right)}\right]+e^{-\left(\gamma_{\mathrm{in}}^{\tau }+\gamma_{\mathrm{out}}^{\tau }\right)\left(t-t_{0}\right)}P_{1}\left(t_{0}\right)\;. \end{array}$$

### 5.3. Energetic Balance

The energy current leaving the reservoir (Equation 12) becomes
(42)$$\begin{array}{cc} I_{E,B}\left(t\right) & =\int d\omega \omega \Gamma \left(\omega \right)\frac{\tau }{2\pi }{\mathrm{sinc}}^{2}\left[\left(\omega -\epsilon \right)\tau /2\right][-\mathrm{Tr}\left\{d^{†}d\rho_{S}\left(t\right)\right\}\left[1-f\left(\omega \right)\right]\\ \; & \;+\mathrm{Tr}\left\{dd^{†}\rho_{S}\left(t\right)\right\}f\left(\omega \right)]\;. \end{array}$$
This current differs from the energy current entering the system (Equation 10)
(43)$$\begin{array}{cc} I_{E,S}\left(t\right) & =\int d\omega \epsilon \Gamma \left(\omega \right)\frac{\tau }{2\pi }{\mathrm{sinc}}^{2}\left[\left(\omega -\epsilon \right)\tau /2\right][-\mathrm{Tr}\left\{d^{†}d\rho_{S}\left(t\right)\right\}\left[1-f\left(\omega \right)\right]\\ \; & \;+\mathrm{Tr}\left\{dd^{†}\rho_{S}\left(t\right)\right\}f\left(\omega \right)]\;, \end{array}$$
and they become equal when τ→∞. When we consider the approximate switching work (Equation 76), we get $\Delta W\left(t_{0}+\tau ,t_{0}\right)\approx \tau \int d\omega \Gamma \left(\omega \right)\left[1-f\left(\omega \right)\right]\mathrm{Tr}\left\{d^{†}d\rho_{S}\left(t_{0}\right)\right\}\left(\omega -\epsilon \right)\frac{\tau }{2\pi }{\mathrm{sinc}}^{2}\left[\left(\omega -\epsilon \right)\tau /2\right]-\tau \int d\omega \Gamma \left(\omega \right)f\left(\omega \right)\mathrm{Tr}\left\{dd^{†}\rho_{S}\left(t_{0}\right)\right\}\left(\omega -\epsilon \right)\frac{\tau }{2\pi }{\mathrm{sinc}}^{2}\left[\left(\omega -\epsilon \right)\tau /2\right]$, where we see that the first law is respected to $O\left\{\Gamma \right\}=O\left\{\lambda^{2}\right\}$.

### 5.4. Entropic Balance

Since the basis diagonalizing the time-dependent density matrix is constant, the energy-resolved rates are constant as well
(44)$$\begin{array}{cc} R_{01,\omega }^{\tau } & =\Gamma \left(+\omega \right)\left[1-f\left(+\omega \right)\right]\frac{\tau }{2\pi }{\mathrm{sinc}}^{2}\left[\left(\omega -\epsilon \right)\tau /2\right]\;,\\ R_{10,\omega }^{\tau } & =\Gamma \left(-\omega \right)f\left(-\omega \right)\frac{\tau }{2\pi }{\mathrm{sinc}}^{2}\left[\left(\omega +\epsilon \right)\tau /2\right]\;, \end{array}$$
and reproduce Equation (Equation 42) when computing the energy current via $I_{E,B}=\int d\omega \omega \sum_{ij}R_{ij,\omega }P_{j}$. We can thus insert the energy current leaving the reservoir (Equation 42) and the matter current $I_{M,B}\left(t\right)=\gamma_{\mathrm{in}}^{\tau }\left[1-P_{1}\left(t\right)\right]-\gamma_{\mathrm{out}}^{\tau }P_{1}\left(t\right)$ into the second law (Equation 23)
(45)$$\begin{array}{cc} {\dot{S}}_{i}^{\tau } & =\left[\gamma_{\mathrm{in}}^{\tau }-\left(\gamma_{\mathrm{in}}^{\tau }+\gamma_{\mathrm{out}}^{\tau }\right)P_{1}\left(t\right)\right]\ln\frac{1-P_{1}\left(t\right)}{P_{1}\left(t\right)}-\beta \left[I_{E,B}\left(t\right)-\mu I_{M,B}\left(t\right)\right]\geq 0\;. \end{array}$$
Here, the first and second contributions of system and reservoir can individually become negative. In fact, in Figure 3 we start from the maximum entropy state in the system, such that the system entropy can only decrease. However, this is then always over-balanced by the other contribution, such that one can see in Figure 3 that the global entropy production is positive. Further, the associated Spohn production rate (Equation 7) still significantly underestimates the global entropy production rate.

One can also see that the global entropy production rate does not vanish for t→∞ as long as τ remains finite (dashed extrapolation of orange curves), in contrast to Spohn’s inequality. This limit t≫τ corresponds to repeated interactions with the reservoir units, and although the system reaches a (nonequilibrium) steady state, the switching work leads to a constant energy current entering the reservoir streams, producing entropy there also at steady state.

## 6. Example: Single Electron Transistor

We have so far discussed examples with an equilibrium environment. The SRL discussed before may directly be extended to two terminals, which in Figure 1 would correspond to two parallel streams of reservoir units, and the dissipator under the weak-coupling assumption decomposes additively in the reservoirs. Then, the expressions for the energy current (Equation 43) can be straightforwardly generalized: The energy current (Equation 13) leaving the reservoir ν becomes
(46)$$\begin{array}{cc} I_{E,B}^{\left(\nu \right)}\left(t\right) & =\int d\omega \omega \Gamma_{\nu }\left(\omega \right)\frac{\tau }{2\pi }{\mathrm{sinc}}^{2}\left[\frac{(\omega -\epsilon )\tau }{2}\right]\left[\left[1-P_{1}\left(t\right)\right]f_{\nu }\left(\omega \right)-P_{1}\left(t\right)\left[1-f_{\nu }\left(\omega \right)\right]\right]\;, \end{array}$$
and similar one gets for the matter current (Equation 22) entering from reservoir ν
(47)$$\begin{array}{cc} I_{M,B}^{\left(\nu \right)}\left(t\right) & =\int d\omega \Gamma_{\nu }\left(\omega \right)\frac{\tau }{2\pi }{\mathrm{sinc}}^{2}\left[\frac{(\omega -\epsilon )\tau }{2}\right]\left[\left[1-P_{1}\left(t\right)\right]f_{\nu }\left(\omega \right)-P_{1}\left(t\right)\left[1-f_{\nu }\left(\omega \right)\right]\right]\;. \end{array}$$

With this, the second law (Equation 23) becomes
(48)$$\begin{array}{c} {\dot{S}}_{i}^{\tau }=\dot{S}-\beta_{L}\left(I_{E,B}^{\left(L\right)}\left(t\right)-\mu_{L}I_{M,B}^{\left(L\right)}\left(t\right)\right)-\beta_{R}\left(I_{E,B}^{\left(R\right)}\left(t\right)-\mu_{R}I_{M,B}^{\left(R\right)}\left(t\right)\right)\geq 0\;. \end{array}$$

We plot the entropy production rate in Figure 4.

There, we see that the BMS entropy production rates of the SET (see also Ref. [49] for this limit) are approached only for comparably large coupling times between system and reservoir (blue).

At steady state (t→∞ but τ finite), the system relaxes to
(49)$$\begin{array}{c} {\bar{P}}_{1}=\frac{\int d\omega \sum_{\nu }\Gamma_{\nu }\left(\omega \right)f_{\nu }\left(\omega \right)\frac{\tau }{2\pi }{\mathrm{sinc}}^{2}\left[\left(\omega -\epsilon \right)\tau /2\right]}{\int d\omega \sum_{\nu }\Gamma_{\nu }\left(\omega \right)\frac{\tau }{2\pi }{\mathrm{sinc}}^{2}\left[\left(\omega -\epsilon \right)\tau /2\right]}\;, \end{array}$$
and accordingly, the system contribution to the second law drops out $\dot{S}\to 0$. Furthermore, we can use that the matter currents at steady state are conserved ${\bar{I}}_{M}\equiv {\bar{I}}_{M,B}^{\left(L\right)}=-{\bar{I}}_{M,B}^{\left(R\right)}$, which allows us to write the second law as ${\dot{S}}_{i}^{\tau }=\left(\beta_{L}\mu_{L}-\beta_{R}\mu_{R}\right){\bar{I}}_{M}-\beta_{L}{\bar{I}}_{E,B}^{\left(L\right)}-\beta_{R}{\bar{I}}_{E,B}^{\left(R\right)}\geq 0$. At steady state, we also have ${\bar{I}}_{E,S}^{\left(L\right)}+{\bar{I}}_{E,S}^{\left(R\right)}=0$, but ${\bar{I}}_{E,B}^{\left(L\right)}+{\bar{I}}_{E,B}^{\left(R\right)}\neq 0$. This implies that, when using expressions for the entropy production rate based on system energy currents, one can for example break the steady-state thermodynamic uncertainty relation [50]. Instead, using our expression for entropy production based on reservoir energy currents, we did numerically not find any violation for multiple parameters.

As an application, we outline how to estimate efficiency bounds following from the second law at steady state. Since at steady state, the system cannot absorb energy anymore, we can write the stationary switching power (Equation 14) simply as ${\bar{P}}_{\mathrm{sw}}=-{\bar{I}}_{E,B}^{\left(L\right)}-{\bar{I}}_{E,B}^{\left(R\right)}$, which allows us to write the second law at steady state as
(50)$$\begin{array}{c} {\bar{\dot{S}}}_{i}^{\tau }=\left(\beta_{L}\mu_{L}-\beta_{R}\mu_{R}\right){\bar{I}}_{M}+\left(\beta_{R}-\beta_{L}\right){\bar{I}}_{E,B}^{\left(L\right)}+\beta_{R}{\bar{P}}_{\mathrm{sw}}\geq 0\;. \end{array}$$
When (without loss of generality) we consider the scenario $\mu_{L}<\mu_{R}$ and $\beta_{L}<\beta_{R}$ (i.e., the left reservoir is hotter $T_{L}>T_{R}$), one can use heat from the hot left reservoir unit to transport electrons through the dot against the potential bias ${\bar{I}}_{M}>0$, generating electric power $P_{\mathrm{el}}=-\left(\mu_{L}-\mu_{R}\right){\bar{I}}_{M}>0$. Considering the original scenario of converting only heat from the hot (left) reservoir to electric power we also assume that the switching power is negative ${\bar{P}}_{\mathrm{sw}}<0$. Then, the efficiency of this process is
(51)$$\begin{array}{cc} \eta & =\frac{-\left(\mu_{L}-\mu_{R}\right){\bar{I}}_{M}}{{\bar{I}}_{E,B}^{\left(L\right)}-\mu_{L}{\bar{I}}_{M}}=\frac{-\left(\mu_{L}-\mu_{R}\right){\bar{I}}_{M}\left(\beta_{R}-\beta_{L}\right)}{\left({\bar{I}}_{E,B}^{\left(L\right)}-\mu_{L}{\bar{I}}_{M}\right)\left(\beta_{R}-\beta_{L}\right)}\\ \; & =\frac{-\left(\mu_{L}-\mu_{R}\right){\bar{I}}_{M}\left(\beta_{R}-\beta_{L}\right)}{\left({\bar{I}}_{E,B}^{\left(L\right)}-\mu_{L}{\bar{I}}_{M}\right)\left(\beta_{R}-\beta_{L}\right)-\beta_{R}\mu_{R}{\bar{I}}_{M}+\beta_{R}{\bar{P}}_{\mathrm{sw}}+\beta_{R}\mu_{R}{\bar{I}}_{M}-\beta_{R}{\bar{P}}_{\mathrm{sw}}}\\ \; & =\frac{-\left(\mu_{L}-\mu_{R}\right){\bar{I}}_{M}\left(\beta_{R}-\beta_{L}\right)}{\left(\beta_{L}\mu_{L}-\beta_{R}\mu_{R}\right){\bar{I}}_{M}+\left(\beta_{R}-\beta_{L}\right){\bar{I}}_{E,B}^{\left(L\right)}+\beta_{R}{\bar{P}}_{\mathrm{sw}}+\beta_{R}\left[-\left(\mu_{L}-\mu_{R}\right){\bar{I}}_{M}-{\bar{P}}_{\mathrm{sw}}\right]}\\ \; & \leq \frac{-\left(\mu_{L}-\mu_{R}\right){\bar{I}}_{M}\left(\beta_{R}-\beta_{L}\right)}{\beta_{R}\left[-\left(\mu_{L}-\mu_{R}\right){\bar{I}}_{M}-{\bar{P}}_{\mathrm{sw}}\right]}\\ \; & =\frac{\beta_{R}-\beta_{L}}{\beta_{R}}\frac{P_{\mathrm{el}}}{P_{\mathrm{el}}-{\bar{P}}_{\mathrm{sw}}}=\eta_{\mathrm{Ca}}\frac{P_{\mathrm{el}}}{P_{\mathrm{el}}-{\bar{P}}_{\mathrm{sw}}}\;. \end{array}$$
Thereby, the wasted switching power reduces the maximum achievable efficiency below the Carnot value.

In contrast to this analysis, continuously operating engines accomplish the conversion of energies while remaining coupled to all reservoirs all the time [51]. Since in these devices one does not have a cost associated to coupling and decoupling processes, they have an intrinsic advantage compared to their finite-stroke counterparts.

## 7. Summary and Conclusions

We have provided a thermodynamic interpretation of the coarse-graining master equation. The switching work required to couple and decouple system and reservoirs leads to a difference between the energy entering the system and the energy leaving the reservoir. With a counting field formalism, we can track the latter and established a second-law inequality, which assumes a standard form despite the fact that the coarse-graining dissipators drag to a nonequilibrium steady state. We exemplified this for the pure-dephasing model, the single resonant level, and the single electron transistor. Although these models are particularly simple and even admit a mostly analytical treatment, we would like to stress that the method can be applied to arbitrary systems. In this case, the time-dependent currents in the second law will have to be calculated numerically. We expect our findings to be relevant for systems that are coupled to reservoirs only for a finite time, e.g., in finite time thermodynamic cycles [52,53,54,55,56,57,58], where the coarse-graining dissipator is a more appropriate choice for finite-time dissipative strokes than the usual BMS limit.

## Figures and Tables

**Figure 1 entropy-22-00525-f001:**
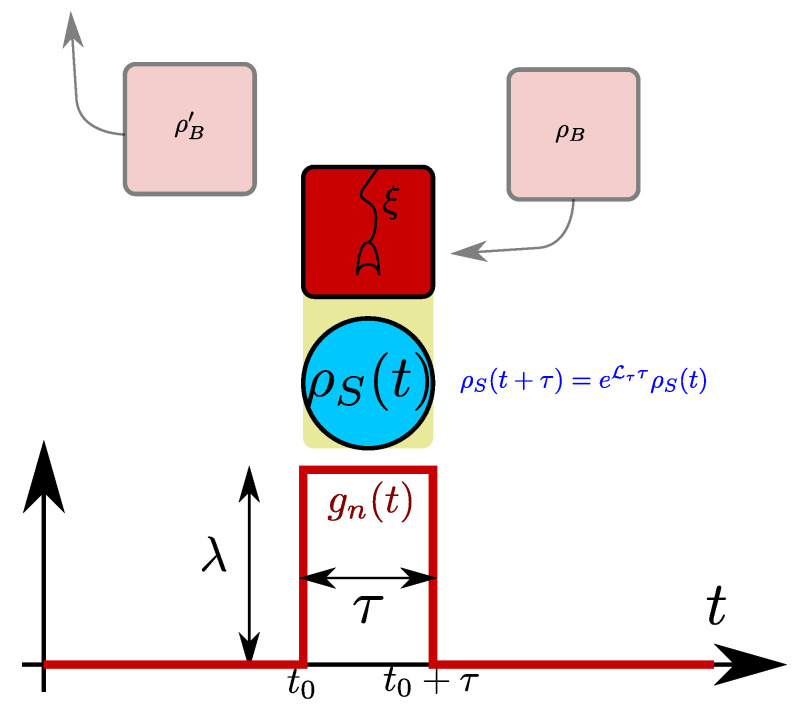
Sketch of the considered setting for a single reservoir. A reservoir unit (top red) is brought into contact with the system (blue) during time $\left[t_{0},t_{0}+\tau \right]$, modeled by a stepwise coupling strength (bottom). After the interaction, another collision can take place with a fresh reservoir, whereas the used reservoir is wasted (faint colors). The effective evolution of the system over the interval $\left[t_{0},t_{0}+\tau \right]$ is described by the coarse-graining dissipator $L_{\tau }$, but the statistics of heat entering the reservoir (detector symbol) can be tracked with a generalized master equation by means of a counting field ξ. The generalization to multiple reservoirs that are coupled simultaneously would induce parallel streams (not shown).

**Figure 2 entropy-22-00525-f002:**
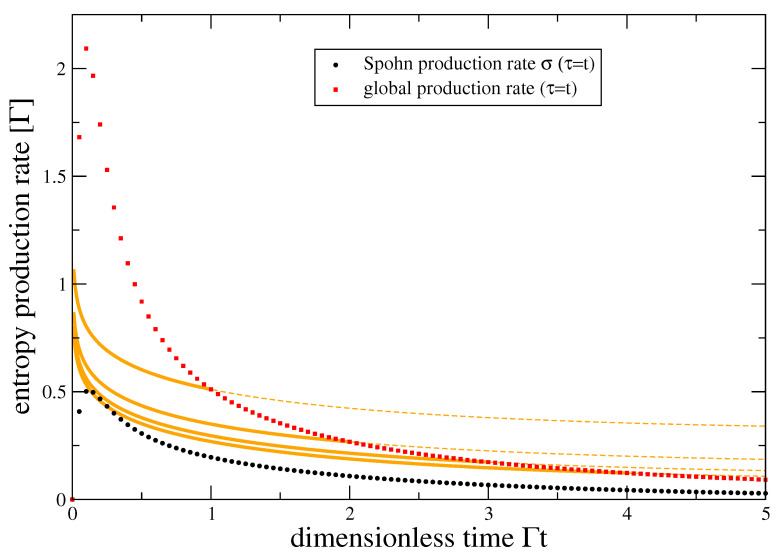
Global entropy production rate ${\dot{S}}_{i}^{\tau }$ for either fixed coarse-graining times (orange, for Γτ∈{1,2,3,4} from top to bottom) or dynamical coarse-graining times (red symbols) and Spohn entropy production rate (black symbols) versus dimensionless time. Bold curve segments correspond to a single unit interaction, whereas the thin dashed projections for t>τ describe repeated system-unit interactions, leading to a finite non-vanishing steady state entropy production rate. Both global entropy production rate and Spohn’s entropy production rate are positive, but the latter underestimates the full entropy production significantly. Parameters: $\rho_{0}^{ij}=1/2$, $\bar{\Gamma }\left(\omega \right)=\Gamma \omega /\omega_{c}e^{-\left|\omega \right|/\omega_{c}}$ with Γβ=1, $\omega_{c}=10\Gamma$.

**Figure 3 entropy-22-00525-f003:**
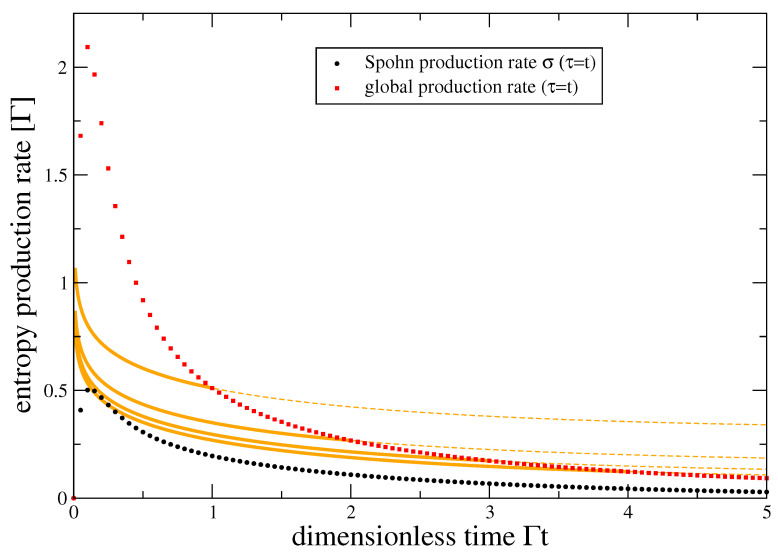
Entropy production rates of the SRL (color coding as in Figure 2). The global entropy production rate (red and orange) is significantly larger than that given by Spohn’s inequality (black). Parameters: $P_{1}\left(0\right)=1/2$, $\Gamma \left(\omega \right)=\Gamma \frac{\delta^{2}}{{\left(\omega -\varepsilon \right)}^{2}+\delta^{2}}$ with ε=ϵ=0, Γβ=0.1, δβ=10, βμ=−2.

**Figure 4 entropy-22-00525-f004:**
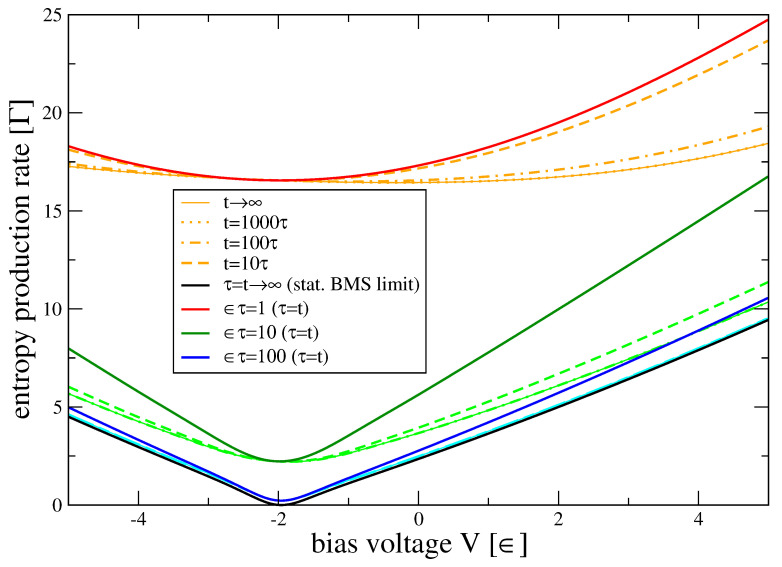
Entropy production rates ${\dot{S}}_{i}^{\tau }\left(t\right)$ of the single electron transistor versus bias voltage $V=\mu_{L}-\mu_{R}$. The stationary BMS entropy production rate (black, t=τ=∞) is finite since the environments are at different thermal equilibrium states, such that a stationary current is flowing, except at its minimum where it vanishes. For finite system and reservoir contact duration τ, the entropy production rate over one contact is significantly larger (red, dark green, dark blue), and in particular does not vanish anywhere as entropy is produced in the reservoirs. This is also observed when the corresponding dissipator is applied repeatedly (light colors with t=10τ (dashed), t=100τ (dash-dotted), and t=1000τ (dotted) and t→∞ (thin solid). Parameters: $P_{1}\left(0\right)=1/2$, $\Gamma_{\nu }\left(\omega \right)=\Gamma_{\nu }\frac{\delta_{\nu }^{2}}{{\left(\omega -\varepsilon_{\nu }\right)}^{2}+\delta_{\nu }^{2}}$ with $\varepsilon_{\nu }=0$, $\Gamma_{\nu }=\Gamma$, $\delta_{\nu }=100\epsilon$, $\Gamma \beta_{L}=0.001$, $\Gamma \beta_{R}=0.1$, $\mu_{L}=+V/2=-\mu_{R}$.

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
