# Peer review of "Thermodynamics of the Coarse-Graining Master Equation"

_entropy, 2020, doi:10.3390/e22050525_

Round 1
Reviewer 1 Report
The paper "Thermodynamics of the Coarse-Graining Master Equation" by G. Schaller and J. Ablaßmayer presents a themodynamic framework valid for the coarse-graning procedures (both statical and dynamical) to derive a master equation in the weak system-reservoir coupling.
In my opinion the paper is well written, the results are interesting and timely, and I do not see any reason which prevents me from recommending this paper for publication in Entropy. However, I suggest the authors to address the following points which are not very clear to me, and so to the potential readers.
1. Lines 36-37. It is written that the Born approximation involves a factorization assumption between system and reservoir. Strictly speaking, if such a factorization exist at the initial time t0, the Born approximation does not require any further assumption on that, because for t>t0 any correlation between system and reservoir is higher order in the perturbation theory. So the Born approximation just refers to the series truncation to the first nontrivial order.
2. Maybe the authors can comment in a sentence or two how the approach should be changed if g_n(t) is not a sharp switching function but something smother.
3. I'm a bit confuse with the possible values of t. For instance, in Eq. (10) or (16) a derivative d/dt is taken, but since t\geq tau, how can you get an infinitesimal dt?
4. Similarly, after Eq. (10) the authors say that t>tau corresponds to repeated system-unit interactions, I guess they mean to say that t=n tau, for integer n, but then t is not a continuous variable...
5. In Eq. (17) second and third line, I guess the notation |j> and |i> are a shortened version of |j(t)> and |i(t)>, right?
6. In Line 199, by why the name "tunneling rate" is used, it is just the spectral density of the fermionic bath, right? Is it a standard name?
8. This is related to the point 2 above: maybe it is good to explain a bit more the sentence "Since continuously operating engines do not require to repeatedly couple and decouple to reservoirs, they have an intrinsic advantage compared to their finite-stroke counterparts."
Author Response
The paper "Thermodynamics of the Coarse-Graining Master Equation" by G. Schaller and J. Ablaßmayer presents a themodynamic framework valid for the coarse-graning procedures (both statical and dynamical) to derive a master equation in the weak system-reservoir coupling.In my opinion the paper is well written, the results are interesting and timely, and I do not see any reason which prevents me from recommending this paper for publication in Entropy. However, I suggest the authors to address the following points which are not very clear to me, and so to the potential readers.
We are very grateful for this positive assessment. Below you find our detailed responses as well as a list of changes.
Lines 36-37. It is written that the Born approximation involves a factorization assumption between system and reservoir. Strictly speaking, if such a factorization exist at the initial time t0, the Born approximation does not require any further assumption on that, because for t>t0 any correlation between system and reservoir is higher order in the perturbation theory. So the Born approximation just refers to the series truncation to the first nontrivial order.
The referee is of course right with this statement. As becomes actually very explicit with the coarse-graining approach, the Born approximation is only required initially or -- in case of repeatedly interacting reservoirs -- at the beginning of the interaction. We have modified this sentence slightly.
Maybe the authors can comment in a sentence or two how the approach should be changed if g_n(t) is not a sharp switching function but something smother.
For a smooth dependence of the coupling functions, calculations evidently become more complicated and fewer analytic advances are possible. Nevertheless, we expect that the spirit of the derivation should still be the same, since due to the vanishing first order expectation values, there are no cross-correlations between successive reservoirs. We have added corresponding remarks in all places where the switching function shows up.
I'm a bit confuse with the possible values of t. For instance, in Eq. (10) or (16) a derivative d/dt is taken, but since t\geq tau, how can you get an infinitesimal dt?
In our derivation, t is just positive. If t-t0<\tau, then we have an approximate solution with a coarse-graining Liouvillian best suited for larger t, if t-t0=tau, we have the optimum, if t-t0\in[n\tau,(n+1)\tau], it corresponds to the situation where the system has interacted with n units and is in the process of interacting with the n+1st. Then, t can become infinitesimally small. in the previous version we were referring to a particular case but we have decided to clarify this now in more detail before Eq. (10).
Similarly, after Eq. (10) the authors say that t>tau corresponds to repeated system-unit interactions, I guess they mean to say that t=n tau, for integer n, but then t is not a continuous variable...
We hope that our clarification before (10) also helps to resolve this remark. If t-t0=n \tau, we have precisely n system-unit interactions, but if t-t0>n \tau and t-t0<(n+1)\tau, we are in the process of interacting with the n+1st unit.
In Eq. (17) second and third line, I guess the notation |j> and |i> are a shortened version of |j(t)> and |i(t)>, right?
True. We have added a remark below (17).
In Line 199, by why the name "tunneling rate" is used, it is just the spectral density of the fermionic bath, right? Is it a standard name?
Yes, the referee is right. We have added the term spectral density. However, in the electronic context this is also called bare tunneling rates.
This is related to the point 2 above: maybe it is good to explain a bit more the sentence "Since continuously operating engines do not require to repeatedly couple and decouple to reservoirs, they have an intrinsic advantage compared to their finite-stroke counterparts."
We have followed this suggestion and added a sentence defining in more detail what we mean by continuously operating engines.
-- list of changes --
numbers without brackets correspond to line numbers, numbers in brackets to Eqns. or Refs, all refer to the revised manuscript
+ 35-43: clarified role of Born approximation in the intro as suggested by referee
+ 59-63: corrected an imprecise a statement on the thermodynamic interpretation (author correction)
+ between (1) and (2): added remark on coupling function time dependence
+ before (10): added clarification on possible values of t vs. \tau
+ after (17): added remark on short-had notation
+ 200: added remark on spectral density vs. bare tunneling rate
+ 238-242: clarified term "continuously operating engines"
+ below (A4): added remark on g_n(t)
+ 263-267: added remarks on smoother coupling function
+ below (A22): carifying remark on the switching process
+ below (A23): carifying remark on the switching process
+ added Refs: [25], [26], [29], [34], [35], [36], [40], [51]
+ updated Ref. [32]
Reviewer 2 Report
In the manuscript "Thermodynamics of the coarse-graining master equation", the authors derive a master equation for systems coupled for a finite time to the environment. The derivation has some similarities to repeated collision models, but the authors consider the system coupled to an infinite reservoir(s). Their formalism also includes full counting statistics to characterize the energy current with the reservoir(s). The authors also carefully check the thermodynamic consistency of the resulting master equation. The paper is well organized and written, with long appendixes with all the technical details. Besides, the proposed formalism is exemplified with simple but illustrative models. This formalism may be used in the study of quantum system coupled for a finite time to a reservoir(s), for example, as the authors suggested, in the analysis of hidden costs in finite-stroke engines. In my opinion, the manuscript could be accepted and published in Entropy.
Author Response
The paper is well organized and written, with long appendixes with all the technical details. Besides, the proposed formalism is exemplified with simple but illustrative models. This formalism may be used in the study of quantum system coupled for a finite time to a reservoir(s), for example, as the authors suggested, in the analysis of hidden costs in finite-stroke engines. In my opinion, the manuscript could be accepted and published in Entropy.
We are very grateful for this positive assessment. The other referee had requested a few minor improvements. Below you also find a list of changes.
-- list of changes --
numbers without brackets correspond to line numbers, numbers in brackets to Eqns. or Refs, all refer to the revised manuscript
+ 35-43: clarified role of Born approximation in the intro as suggested by referee
+ 59-63: corrected an imprecise a statement on the thermodynamic interpretation (author correction)
+ between (1) and (2): added remark on coupling function time dependence
+ before (10): added clarification on possible values of t vs. \tau
+ after (17): added remark on short-had notation
+ 200: added remark on spectral density vs. bare tunneling rate
+ 238-242: clarified term "continuously operating engines"
+ below (A4): added remark on g_n(t)
+ 263-267: added remarks on smoother coupling function
+ below (A22): carifying remark on the switching process
+ below (A23): carifying remark on the switching process
+ added Refs: [25], [26], [29], [34], [35], [36], [40], [51]
+ updated Ref. [32]
Round 2
Reviewer 1 Report
After the revisions made by the authors, I recommend this manuscript for publication in Entropy.